



# Hierarchical scale dependence associated with the extension of the nonlinear feedback loop in a seven-dimensional Lorenz model

B.-W. Shen[1]

[1] Dr. Bo-Wen Shen
Department of Mathematics and Statistics
San Diego State University
5500 Campanile Drive
San Diego, CA 92182-7720.
E-mail: bshen@mail.sdsu.edu

*Correspondence to:* Dr. Bo-Wen Shen
(bshen@mail.sdsu.edu; bowen.shen@gmail.com)



**Abstract.** In this study, we construct a seven-dimensional Lorenz Model (7DLM) to discuss the impact of an extended nonlinear feedback loop on solutions' stability and illustrate the hierarchical scale dependence of chaotic solutions. Compared to the 5DLM, the 7DLM includes two additional high wavenumber modes that are selected based on an analysis of the nonlinear temperature advection term, a Jacobian term $(J(\psi, \theta))$, where, $\psi$ and $\theta$ represent the streamfunction and temperature perturbations, respectively. Fourier modes that represent temperature in the 7DLM can be categorized into three major scales as the primary (the largest scale), secondary, and tertiary (the smallest scale) modes. Further extension of the nonlinear feedback loop within the 7DLM can provide negative nonlinear feedback to stabilize solutions, thus leading to a much larger critical value for the Rayleigh parameter (rc $\sim$ 116.9) for the onset of chaos, as compared to an rc of 42.9 for the 5DLM as well as an rc of 24.74 for the 3DLM. The rc is determined by an analysis of ensemble Lyapunov exponents (eLEs) with Prandtl number ($\sigma$) of 10. To examine the dependence of rc on the value of the Prandtl number, a linear stability analysis is performed near the nontrivial critical point using a wide range of Rayleigh parameter ($40 \leq r \leq 195$) and Prandtl number ($5 \leq \sigma \leq 25$). Then an eLE analysis is conducted using selected values of the Prandtl number. The linear stability analysis is done by solving for the analytical solutions of the critical points, by linearizing the 7DLM with respect to the analytical solutions, and by calculating the eigenvalues of the linearized system. Within the range of ($5 \leq \sigma \leq 25$), the 7DLM requires a larger $rc$ for the onset of chaos than the 5DLM.

In addition to the negative nonlinear feedback illustrated and emulated by the quasi-equilibrium state solutions for high wavenumber modes, the 7DLM reveals the hierarchical scale dependence of chaotic solutions. For chaotic solutions with (r=120), the Pearson correlation coefficients (PCCs) between the primary and secondary modes (i.e., $Z$ and $Z_1$) and between the secondary and tertiary modes (i.e., $Z_1$ and $Z_2$) are 0.988 and 0.998, respectively. Here, $Z$, $Z_1$, and $Z_2$ represent the time-varying amplitudes of the primary, secondary, and tertiary modes, respectively. High PCCs indicate a strong linear relationship among the modes at various scales and a hierarchy of scale dependence. Future work will be undertaken to examine how higher dimensional LMs may produce a larger critical value for the Rayleigh parameter for the onset of chaos and reveal stronger hierarchical scale dependence.

## 1 Introduction

In 1963, Prof. Lorenz of MIT published two important papers that first introduced the concept of finite predictability using an idealized model that contained three ordinary differential equations (Lorenz, 1963a), and classified three kinds of predictability (Lorenz, 1963b). The idealized model, derived from the nonlinear partial differential equations governing Rayleigh-



Benard convection, is known as the three-dimensional Lorenz model (3DLM). The 3DLM
was used to illustrate the sensitive dependence of numerical solutions on tiny changes in ini-
tial conditions. Appearing only in nonlinear models, this unique feature is known as chaos or
the butterfly effect (e.g., Gleick, 1987; IPCC, 2007; Anthes, 2011). In our previous studies,

this feature is referred to as the butterfly effect of the first kind (e.g., Shen 2014a, 2015b).
Based on the numerical phenomenon, it has been inferred that tiny perturbation may alter
the large-scale flow (e.g., producing a tornado in Lorenz, 1972), which is referred to as the
the butterfly effect of the first kind (e.g., Pielke 2008; Shen 2014a). However, it has been
suggested that the appearance of a butterfly effect of the first kind does not directly lead

to a butterfly effect of the second kind. Such a suggestion is due to the fact that although
a butterfly effect of the first kind may appear within a numerical model with a finite de-
gree of nonlinearity (e.g., the original Lorenz model), an improved degree of nonlinearity in
high-order Lorenz models (e.g., Shen 2014a; 2015b) can mitigate or suppress the sensitive
dependence of simulations on initial conditions. In the second important paper in 1963,

Lorenz categorized predictability as either intrinsic predictability, attainable predictability
or practical predictability. These three types of predictability show dependence on the na-
ture of flows at various scales, the accuracy of initial conditions, and the mathematical
formula of the numerical models. The first and second publications of Lorenz in 1963 indeed
suggested the dependence of predictability on models (as well as initial conditions), implying

the dependence of chaotic solutions on models.

    Lorenz's studies have made significant influences on the activities of both real-world mod-
els and idealized models. To minimize the negative impact of inaccurate initial conditions,
sophisticated data assimilation schemes and systems have been developed in order to im-
prove forecasts. On the other hand, high-order Lorenz models (e.g., Curry, 1978; Curry

et al., 1984; Franceschini and Tebaldi, 1985; Howard and Krishnamurti, 1986; Franceschini
et al., 1988; Hermiz et al., 1995; Kennamer, 1995; Thiffeault and Horton, 1996; Musielak
et al., 2005; Chen and Price, 2006; Roy and Musielak, 2007a,b,c; Lucarini and Fraedrich,
2009) and high-resolution real-world models (e.g., Shen et al., 2006a; 2012; 2013) have been
developed in order to understand the impact of increasing resolutions that may improve

systems' stability due to the improved accuracy of model formulas. Using high-order Lorenz
models with a finite number of modes, a different mode truncation has been shown to impact
the stability of numerical solutions. Earlier studies led to an inconclusive conclusion as to
whether increasing the number of modes can produce a model with better predictability.

    Recent studies (Shen 2014a; 2015b) based on the five- and six-dimensional Lorenz models

(5D and 6D LMs) indicated that selections of high wavenumber modes that can properly
extend the nonlinear feedback loop of the original 3D Lorenz model may produce a negative
nonlinear feedback to stabilize solutions. Furthermore, the impact of the negative nonlinear





feedback was illustrated using revised 3DLMs with one or two parameterized terms that can emulate the effect of the negative feedback (e.g., Shen 2014a; 2015a). The 5DLM and
6DLM, as well as the revised 3DLMs with parameterizations, require a larger value for the normalized Rayleigh parameter for the onset of chaos. In addition to the negative nonlinear feedback that comes from the nonlinear terms and dissipative terms in association with newly added modes, the destabilizing impact (i.e., positive feedback) of an additional heating term that appears in the 6DLM has been identified and examined by comparing it with the
5DLM. Studies based on the 5D and 6DLM collectively suggest that the various roles of newly resolved small-scale processes can either stabilize or destabilize solutions, consistent with the impact of butterfly effect as stated by Lorenz (1972). Therefore, in general, to understand the impact of newly added high wavenumber modes on solution stability, it is important to examine the competing/collective impact of small-scale processes. The major
similarities and differences between the 5DLM and 6DLM are as follows: (1) both models include negative nonlinear feedback that is associated with the extended nonlinear feedback loop; (2) the 6DLM includes an additional high wavenumber streamfunction mode that introduces an additional time dependent equation for its amplitude, an additional heating term, and several nonlinear terms. To improve the stability of high-dimensional LMs, based
on the studies with the 5DLM and 6DLM, we suggested that it is important to select new modes to extend the nonlinear feedback loop that can effectively provide negative nonlinear feedback and that it is not critical to include an additional (streamfunction) mode that leads to an additional heating term to provide a positive feedback. Therefore, in this study, we construct a 7DLM using an approach similar to the 5DLM that extends the nonlinear
feedback loop without introducing a new heating term.

The long-term goal is to determine under what conditions increasing resolutions can improve the predictions in weather/climate models. To achieve this goal, we first derive the higher-dimensional Lorenz models in order to illustrate the impact of the newly resolved small processes and the additional nonlinear terms (associated with the various mode trun-
cations and model coupling) on system stability. Additionally, the high-dimensional LMs and the revised 3DLM with parameterized terms can be used to test the performance of the numerical methods in calculating the Lyapunov exponents (LEs). These types of studies may help identify an appropriate method for the LE calculation in real world models. Then, the impact of small-scale processes, resolved by new changes in a model, on the solution
stability can be better examined. The paper is organized as follows. Section 2 discusses the governing equations, the seven-dimensional Lorenz model, and the revised three-dimensional Lorenz model with parameterizations, analytical solutions for non-trivial critical points, and numerical approaches. Discussions of numerical solutions and analytical solutions are provided in section 3. A conclusion is provided at the end. Detailed discussion regarding



the selection of new modes based on the analysis of the Jacobian term is provided in the Supplemental Materials of Shen (2015b).

## 2 The seven-dimensional Lorenz model and numerical approaches

### 2.1 The governing equations

The governing equation for 2D (x,z), incompressible, and Boussinesq flow can be written as:

$$\frac{\partial}{\partial t}\nabla^2\psi = -J(\psi,\nabla^2\psi) + \nu\nabla^4\psi + g\alpha\frac{\partial\theta}{\partial x}, \qquad (1)$$

$$\frac{\partial\theta}{\partial t} = -J(\psi,\theta) + \frac{\Delta T}{H}\frac{\partial\psi}{\partial x} + \kappa\nabla^2\theta, \qquad (2)$$

where $\psi$ is the streamfunction that yields the $u = -\psi_z$ and $w = \psi_x$, which, respectively, represent the horizontal and vertical velocities; $\theta$ is the temperature perturbation; and $\Delta T$ is the temperature difference at the bottom and top boundaries. The constants, $g$, $\alpha$, $\nu$, and $\kappa$ denote the acceleration of gravity, the coefficient of thermal expansion, the kinematic viscosity, and the thermal conductivity, respectively. The Jacobian of two arbitrary functions is defined as $J(A,B) = (\partial A/\partial x)(\partial B/\partial z) - (\partial A/\partial z)(\partial B/\partial x)$. Additionally, $\nabla^4\psi = \partial/\partial x(\nabla^2\partial\psi/\partial x) + \partial/\partial z(\nabla^2\partial\psi/\partial z)$. The above equations were used by Saltzman (1962) to study finite amplitude convection. Later, Lorenz (1963a) used them to derive the 3DLM.

### 2.2 The 7D Lorenz Model (7DLM)

In this section, we discuss how the 7DLM is constructed using the following seven Fourier modes:

$$M_1 = \sqrt{2}sin(lx)sin(mz), \quad M_2 = \sqrt{2}cos(lx)sin(mz), \quad M_3 = sin(2mz), \qquad (3)$$

$$M_5 = \sqrt{2}cos(lx)sin(3mz), \quad M_6 = sin(4mz), \qquad (4)$$

$$M_8 = \sqrt{2}cos(lx)sin(5mz), \quad M_9 = sin(6mz). \qquad (5)$$

Here the horizontal and vertical wavenumbers ($l$ and $m$) are defined as $\pi a/H$ and $\pi/H$, respectively. The parameter $a$ is defined as the ratio of the vertical scale of the convection cell to its horizontal scale (i.e., $a = l/m$). The term $H$ is the domain height, and $2H/a$ indicates the domain width. Using these modes, $\psi$ and $\theta$ can be represented as follows:

$$\psi = C_1\bigg(XM_1\bigg), \qquad (6)$$

$$\theta = C_2\bigg(YM_2 + Y_1M_5 + Y_2M_8 - ZM_3 - Z_1M_6 - Z_2M_9\bigg), \qquad (7)$$

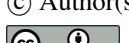

$$C_1 = \kappa \frac{(1+a^2)}{a}, \quad C_2 = \frac{\Delta T}{\pi} \frac{R_c}{R_a}, \quad R_c = \frac{\pi^4}{a^2}(1+a^2)^3, \quad R_a^{-1} = \frac{\nu \kappa}{g \alpha H^3 \Delta T},$$

where $C_1$ and $C_2$ are constants, $R_a$ is the Rayleigh number, and $R_c$ is its critical value for the free-slip Rayleigh-Benard problem. Using Eqs. (6-7), solutions within the 7DLM are

represented by the seven spatial modes $M_1 - M_3$, $M_5 - M_6$ and $M_8 - M_9$, as well as their corresponding time-varying amplitudes $(X, Y, Z, Y_1, Z_1, Y_2, Z_2)$, respectively. By comparison, Eq. (3) was used to derive the 3DLM and Eqs. (3-4) were utilized to construct the 5DLM. While the 3DLM, 5DLM, and 7DLM have one horizontal wavenumber, they contain two, four, and six vertical wavenumbers, respectively. To facilitate discussion, in the text below,

$M_1$ is referred to as primary streamfunction mode, $M_2$ and $M_3$ are referred to as primary temperature modes, $M_5$ and $M_6$ are referred to as secondary temperature modes, and $M_8$ and $M_9$ are referred to as tertiary temperature modes. While Shen (2014a) derived the 5DLM in order to discuss the impact of the secondary temperature modes (i.e., $Y_1$ and $Z_1$) on solution stability, here, we construct the 7DLM in order to examine the impact of the

tertiary temperature modes (i.e., $Y_2$ and $Z_2$) as compared to the primary and secondary temperature modes. We additionally discuss how the nonlinear feedback associated with the secondary and tertiary modes can be emulated using a parameterized term that can be added into the 3DLM to improve stability.

Note that Shen (2015a) extended the 5DLM into a 6DLM by including the secondary

streamfunction mode (i.e., $M_4 = \sqrt{2} sin(lx) sin(3mz)$) and examined its impact on solution stability. The author found that the additional streamfunction mode introduces an additional heating term that can destabilize the solution. However, since the negative nonlinear feedback by the secondary temperature modes dominates, the critical value of Rayleigh parameter ($rc \sim 41.1$) for the 6DLM is slightly smaller than that of the 5DLM ($rc \sim 42.9$).

Similarly, using a 9DLM that includes the secondary and tertiary "streamfunction" modes (i.e., $M_4$ and $M_7 = \sqrt{2} sin(lx) sin(5mz)$) as well as the tertiary temperature modes ($M_8$ and $M_9$), it can be shown that $M_7$ also introduces an additional heating term that provides positive feedback. The 9DLM is a superset of the 7DLM, but the 7DLM is not a superset of the 6DLM because the 7DLM does not include the secondary or tertiary streamfunction

mode. Since the analytical solutions of the critical points for the 7DLM can be obtained to perform linear stability analysis, and since the tertiary temperature modes produce a stronger feedback than the tertiary streamfunction modes in the 9DLM, in this study, we will simply discuss the 7DLM.

To transform the partial differential equations (Eqs. 1-2) into a set of ordinary differential

equations, a major step is representing the nonlinear Jacobin terms using the Fourier modes. As discussed in Shen (2014a; 2015b), the original Lorenz model only includes nonlinear terms



from the the Jacobian term of Eq. (2), which is written as follows:

$$J(\psi,\theta) = C_1 C_2 \Bigg( XYJ(M_1, M_2) - XZJ(M_1, M_3) + XY_1 J(M_1, M_5) - XZ_1 J(M_1, M_6)$$

$$+ XY_2 J(M_1, M_8) - XZ_2 J(M_1, M_9) \Bigg). \tag{8}$$

Note that the 3DLM only contains the first two terms on the right hand side of Eq. (8), namely, $XYJ(M_1, M_2)$ and $-XZJ(M_1, M_3)$. These two terms form the nonlinear feedback loop within the 3DLM. The nonlinear feedback loop is extended in the 5DLM that includes the first four terms. The nonlinear feedback loop within the 3DLM may be viewed as the main trunk and its extension in the 5DLM as a branch. Detailed discussion regarding the nonlinear feedback loop of the 3DLM and its extension in the 5DLM can be found in Shen (2014a) and the Supplemental Materials of Shen (2015b). As discussed below, in this study, modes $M_8$ and $M_9$ are selected to further extend the nonlinear feedback loop. Based on the analysis of the Jacobian, we have:

$$J(M_1, M_6) \approx \sqrt{2} m l cos(lx)(2sim(5mz) + 2sim(-3mz)) = 2ml(M_8 - M_5). \tag{9}$$

In Eq. (9), the mode of $sin(5mz)$ (i.e., $M_8$) is required as a result of the interaction between $M_1$ and $M_6$ modes (i.e., $J(M_1, M_6)$). As shown in Figure 1, the downscale transfer associated with the $M_8$ mode is disabled in the 5DLM because the $M_8$ is ignored. With the $M_8$ mode, we have:

$$J(M_1, M_8) \approx ml(3sim(6mz) + 2sim(-4mz)) = ml(3M_9 - 2M_6), \tag{10a}$$

which requires a mode of $sin(6mz)$ (i.e., $M_9$) in order to enable a downscale transfer from the interaction of $M_1$ and $M_8$ via $J(M_1, M_8)$. The inclusion of $M_9$ leads to:

$$J(M_1, M_9) \approx \sqrt{2} m l cos(lx)(3sim(-5mz)) = -3mlM_8, \tag{10b}$$

which provides a feedback to the $M_8$ mode via an upscale transfer. Equations (10a) and (10b) form a new feedback loop $(M_8 - M_9 - M_8)$ that is an extension (or branch) of the extended nonlinear feedback loop in the 5DLM.

After derivations, the following seven equations of the 7DLM can be obtained:

$$\frac{dX}{d\tau} = -\sigma X + \sigma Y, \tag{11}$$

$$\frac{dY}{d\tau} = -XZ + rX - Y, \tag{12}$$

$$\frac{dZ}{d\tau} = XY - XY_1 - bZ, \tag{13}$$

$$\frac{dY_1}{d\tau} = XZ - 2XZ_1 - d_o Y_1, \tag{14}$$





$$\frac{dZ_1}{d\tau} = 2XY_1 - 2XY_2 - 4bZ_1, \tag{15}$$

$$\frac{dY_2}{d\tau} = 2XZ_1 - 3XZ_2 - d_1Y_2, \tag{16}$$

$$\frac{dZ_2}{d\tau} = 3XY_2 - 9bZ_2. \tag{17}$$

Here, $\tau = \kappa(1+a^2)(\pi/H)^2 t$ (dimensionless time), $\sigma = \nu/\kappa$ (the Prandtl number), $r = R_a/R_c$ (the normalized Rayleigh number or the heating parameter), $b = 4/(1+a^2)$, $d_o = (9+a^2)/(1+a^2)$, and $d_1 = (25+a^2)/(1+a^2)$.

When the tertiary mode $(Y_2, Z_2)$ are neglected, the 7DLM is reduced to become the 5DLM and becomes the 3DLM when secondary and tertiary modes $(Y_1, Z_1, Y_2, Z_2)$ are ignored. Alternatively, Eqs. (11-15) can be viewed as a 5DLM with feedback processes that result from the tertiary modes (i.e., $-2XY_2$ in Eq. 15); Eqs. (11-13) can be viewed as a 3DLM with feedback processes that arise from the secondary and tertiary additional modes (i.e., $-XY_1$ in Eq. 13). The role of $XY_1$ will be discussed below in greater detail. Here, and in Shen (2014a), unless otherwise stated, the term "feedback" refers to a nonlinear process that involves the secondary modes, namely $(Y_1,$ and/or $Z_1)$, or the tertiary modes, $(Y_2$ and $Z_2)$.

## 2.3 Energy Conservation in the 7D non-dissipative LM

Using Eqs. (14-15) of Shen (2015b) for definitions of domain-averaged kinetic energy $(\overline{KE})$, available potential energy $(\overline{APE})$, and potential energy $(\overline{PE})$, (e.g., Treve and Manley, 1982; Thiffeault and Horton, 1996; Blender and Lucarini, 2013; Shen, 2014a), we obtain the following equations:

$$\overline{KE} = \frac{C_o}{2}\left(X^2\right), \tag{18}$$

$$\overline{APE} = -\frac{C_o}{2}\frac{\sigma}{r}\left(Y^2 + Z^2 + Y_1^2 + Z_1^2 + Y_2^2 + Z_2^2\right), \tag{19}$$

$$\overline{PE} = -C_o\sigma\left(Z + Z_1/2 + Z_2/3\right), \tag{20}$$

where $C_o = \pi^2\kappa^2(\frac{1+a^2}{a})^3$. Equations (18) and (19) yield the following:

$$\overline{KE} + \overline{APE} = \frac{C_o}{2}\left(X^2 - \frac{\sigma}{r}(Y^2 + Z^2 + Y_1^2 + Z_1^2 + Y_2^2 + Z_2^2)\right) = C_3, \tag{21}$$

while Eqs. (18) and (20) lead to the following:

$$\overline{KE} + \overline{PE} = C_o\left(\frac{X^2}{2} - \sigma(Z + \frac{Z_1}{2} + \frac{Z_2}{3})\right) = C_4. \tag{22}$$



With Eqs. (11-17) in the dissipationless limit, the time derivative of both Eqs. (21) and
(22) are zero, so both $C_3$ and $C_4$ are constants. Therefore, Eqs. (21-22) indicate two energy
conservation laws. The above analysis also suggests that when $Y_2$ (or $Y_1$) is included, $Z_2$ (or
$Z_1$) should be included in order to conserve the energy in the dissipationless limit. Thus,
both $Y_2$ and $Z_2$ (or $Y_1$ and $Z_1$) are viewed as the tertiary (or secondary) modes.

With Eq. (22) and the 7DLM (Eqs. 11-17), the time derivative of the total energy
becomes:

$$\frac{d\overline{KE}}{d\tau} + \frac{d\overline{PE}}{d\tau} = C_3\sigma\left(-X^2 + bZ + 2bZ_1 + 3bZ_2\right), \tag{23}$$

which gives steady state solutions as follows:

$$X_c = \pm\sqrt{b(Z_c + 2Z_{1c} + 3Z_{2c})}, \tag{24}$$

where, a subscript 'c' indicates a critical point solution that will be discussed in details
in Section 2.5. Equation (24) can be used to verify the solutions solved by the analytical
method or numerical method.

### 235  2.4 A revised 3DLM with a parameterized term using the 7DLM

In this section, we discuss how the nonlinear feedback processes resolved by high wavenumber
modes in the 7DLM can be emulated into a revised 3DLM. Throughout the discussion, the
negative nonlinear feedback provided by the high wavenumber modes will be illustrated. The
basic idea is to express higher wavenumber modes in terms of lower wavenumber modes. To
achieve this, two steps are required. First, an assumption of quasi equilibrium state for the
tertiary modes is made in order to express the tertiary modes in terms of the primary and
secondary modes. Simply speaking, an assumption is made that $dY_2/d\tau \approx 0$ and $dZ_2/d\tau \approx 0$
in Eqs. (16-17) and $Y_2$ and $Z_2$ are solved as follows:

$$Y_2 = \frac{2bXZ_1}{X^2 + bd_1}, \tag{25}$$

$$Z_2 = \frac{2}{3}\frac{X^2 Z_1}{X^2 + bd_1}. \tag{26}$$

Equations (25-26) are referred to as quasi-equilibrium state solutions or the "steady-state"
solutions. Eq. (25) suggests that $2XY_2$ is proportional to $4bZ_1$ and may effectively act as
an additional dissipation term similar to the third term $4bZ_1$ on the right hand side of Eq.
(15). In the second step, by plugging Eq. (25) into Eq. (15), the quasi equilibrium state
solutions of secondary modes ($Y_1$ and $Z_1$ of Eqs. 14-15) can be represented in terms of the
primary modes, as follows:

$$Y_1 = \frac{bXZ\frac{2X^2+bd_1}{X^2+bd_1}}{X^2 + bd_0\frac{2X^2+bd_1}{X^2+bd_1}}, \tag{27}$$





$$Z_1 = \frac{1}{2} \frac{X^2 Z}{X^2 + bd_0 \frac{2X^2 + bd_1}{X^2 + bd_1}}. \tag{28}$$

Using Eq. (27), $XY_1$ in Eq. (13) can be represented in terms of the primary modes, and is added into a revised 3DLM as follows:

$$\frac{dX}{d\tau} = -\sigma X + \sigma Y, \tag{29}$$


$$\frac{dY}{d\tau} = -XZ + rX - Y, \tag{30}$$

$$\frac{dZ}{d\tau} = XY - bZ + Q_2, \tag{31}$$

$$Q_2 = -XY_1 = \frac{-bX^2 Z \frac{2X^2 + bd_1}{X^2 + bd_1}}{X^2 + bd_0 \frac{2X^2 + bd_1}{X^2 + bd_1}}. \tag{32}$$

Although $Q_2$ in Eq. (32) is a function of the primary modes $(X, Y, Z)$, it is indeed proportional to $-2bZ$, which may effectively play a role similar to $-bZ$ in Eq. (31). The

nonlinear feedback process (by $-XY_1$), which is explicitly resolved by the secondary and tertiary modes in the 7DLM, is "emulated" (or parameterized) by the primary modes. Thus, Eqs. (29-31) with the $Q_2$ term, as defined in Eq. (32), are referred to as the 3DLM-P7d or 3DLMP7d, which indicates a revised 3DLM with a parameterized term using the 7DLM. Previously, Shen (2015a) discussed the revised 3DLMs with a parameterized term using the

5DLM (i.e., 3DLM-P5d) and 6DLM (i.e., 3DLM-P6d). Following the discussion in Shen (2015a), it can be shown that the critical points in the 3DLM-P7d are the same as those in the 7DLM. The critical point solutions of the 7DLM are discussed in the next section. In comparison to the Q2 terms of the 3DLM-P7d (Eq. 32) and the 3DLM-P5d (Eq. 10 of Shen 2015a), it should be noted that the former cannot be reduced to be the latter by simply

assuming $d_1 = 0$. This is because via nonlinear terms, $Y_2$ and $Z_2$ still provide feedback to both the secondary modes and the primary modes when $d_1 = 0$. When $X$ is relatively small as compared to $bd_0$ and $bd_1$, Eq. (32) can be simplified as $Q_2 = -qX^2$ and $q$ a non-negative number when $Z$ is positive. When $X$ is very large as compared to $bd_0$ and $bd_1$, Eq. (32) can be simplified to $Q_2 \approx -2bZ$. Under the same condition, Eqs. (28) and (26) lead to

$Z_1 \approx Z/2$ and $Z_2 \approx 2Z_1/3 \approx Z/3$, respectively. These constraints and Eqs. (24) lead to $X \approx \sqrt{3bZ}$ under the condition of $X >> bd_0$ and $X >> bd_1$.

## 2.5 Analytical solutions of critical points in the 7DLM

Here, to examine the linear stability, the analytical solutions of critical points in the 7DLM are presented. Plugging Eq. (27) into Eq. (13), the seven, time-independent counterpart of Eq. (11-17) can be reduced to become one, time-independent equation for the critical point solution of $X$ (i.e., $X_c$):

$$X_c^6 + BX_c^4 + CX_c^2 + D = 0, \tag{33}$$


where

$$B = 2bd_0 + bd_1 - 3bZ_c, \tag{34a}$$

$$C = b^2 d_0 d_1 - (2b^2 d_0 + 2b^2 d_1)Z_c, \tag{34b}$$

$$D = -b^3 d_0 d_1 Z_c. \tag{34c}$$

As discussed below, $Z_c$ is equal to $r-1$. By assuming $P = X_c^2$, Eqs. (33-34) become the well-known cubic equation, and its analytical solutions can be expressed using the cubic formula.

The procedures of Press et al. (1992) are followed in order to solve for $P$, which yields $X_c = \pm\sqrt{P}$. From Eqs. (11-12), we can obtain $Y_c = X_c$ and $Z_c = r-1$. As the critical point solutions for the primary modes are obtained, the critical point solutions for the secondary and tertiary modes in Eqs. 25-28 are also determined. Thus, we have the following critical point solutions for the 7DLM:

$$X_c = \pm\sqrt{P}, \tag{35a}$$

$$Y_c = X_c, \tag{35b}$$

$$Z_c = r - 1, \tag{35c}$$

$$Y_{1c} = \frac{bX_cZ_c\frac{2X_c^2+bd_1}{X_c^2+bd_1}}{X_c^2 + bd_0\frac{2X_c^2+bd_1}{X_c^2+bd_1}}, \tag{35d}$$

$$Z_{1c} = \frac{1}{2}\frac{X_c^2 Z_c}{X_c^2 + bd_0\frac{2X_c^2+bd_1}{X_c^2+bd_1}}, \tag{35e}$$

$$Y_{2c} = \frac{2bX_cZ_{1c}}{X_c^2 + bd_1}, \tag{35f}$$

$$Z_{2c} = \frac{2}{3}\frac{X_c^2 Z_{1c}}{X_c^2 + bd_1}. \tag{35g}$$

**2.6 Numerical approaches**

To obtain numerical solutions for Lorenz models, Fortran codes were previously developed based on the implementation of the 4th order Runge-Kutta scheme (e.g., Shen, 2014a). The codes are modified and used for calculation of ensemble Lyapunov exponents (e.g., Benettin et al., 1980) in the 7DLM, as discussed in Figure 2 below. A variety of numerical

and statistical packages in R (e.g., Adler, 2012) are additionally applied to obtain numerical solutions for the Lorenz models and to perform analysis of model solutions. Unless otherwise stated, Fortran codes and smaller time interval (e.g., 0.0001) are used to obtain results for better accuracy (e.g., Figure 2), and R codes with larger time interval (e.g., 0.01) are used to obtain numerical results for comparisons (which are provided in most figures alongside

numerical solutions obtained from the Lorenz models.) With the exception for the heating parameter ($r$) and the Prandtl number ($\sigma$), the following parameters are kept as constant:





including $a = 1/\sqrt{2}$, $b = 8/3$, $d_o = 19/3$, $d_1 = 17$, and a minimum value for $R_c = 27\pi^4/4$. A value of $\sigma = 10$ is used in most cases, although additional values are examined to determine their impact on solution stability, as shown in Figure 5 To outline the characteristics of
solutions with no loss of generality, in Figs. 3, 6, and 8-11,, the initial conditions are as follows:

$$(X, Y, Z, X_1, Y_1, Z_1, Y_2, Z_2) = (0, 1, -1, 0, 0, 0, 0, 0). \tag{36a}$$

In Figure 7, the initial conditions are given at one non-trivial critical point with the same perturbation in $Y$ and $Z$, yielding

$$(X, Y, Z, X_1, Y_1, Z_1, Y_2, Z_2) = (X_c, Y_c + 1, Z_c - 1, Y_{1c}, Z_{1c}, Y_{2c}, Z_{2c}). \tag{36b}$$

Note that the initial condition with a nonzero $Z$ is slightly different from the one in previous studies (Shen 2014a and 2015a,b). The change is made to illustrate an $rc$ of 116.9, which is consistent with the calculation of the ensemble Lyapunov exponent analysis, as discussed below. The change does not lead to a different conclusion. In Figures 3, 6 and 8-10, the
dimensionless time interval ($\triangle \tau$) is 0.01 and the total number of time steps ($N$) is 10,000, yielding a total dimensionless time ($\tau$) of 100. In Figures 7 and 11, a larger number of time steps is used. The former uses $N = 180,000$ while the latter has $N = 100,000$. To better compare the solutions obtained from different LMs in Figure 3, the results are rescaled using the analytical solutions of the critical points, [i.e., Eq. 21 and Eq. 19 of Shen (2014a) for
the 3DLM and 5DLM, respectively; and Eq. 35 for the 7DLM].

To quantitatively evaluate whether or not the system is chaotic, we calculate the Lyapunov exponent (LE, e.g., Benettin et al., 1980; Froyland and Alfsen, 1984; Wolf et al., 1985; Nese, 1989; Zeng et al. 1991; Eckhardr and Yao, 1993; Christiansen and Rugh, 1997; Kazantsev 1999; Sprott, 1997, 2003; Ding and Li, 2007; Li and Ding, 2011) using the trajectory separa-
tion (TS) method (Sprott, 1997, 2003) and the Gram–Schmidt reorthonormalization (GSR) method (Wolf et al., 1985; Shen, 2014a and references therein). Using the given initial conditions (ICs) and a set of parameters in the LMs, the TS scheme calculates the largest LE and the GSR scheme produces "n" LEs where "n" is the dimension of the 5D or 7D LM. For the LE analysis, we use the Fortran codes with $\triangle \tau = 0.0001$ and $N = 10,000,000$ that yields
$\tau = 1,000$. To minimize dependence on the ICs, 10,000 ensemble ($En$=10,000) runs with the same model configurations but different ICs are performed, and an ensemble averaged LE (eLE) is obtained from the average of the 10,000 LEs. A large $N$ and $En$ are used to understand the long-term average behavior for the LM solutions. Figure A1 in Appendix A shows the initial conditions for the ensemble runs that represent white Gaussian noises.

As compared to earlier studies, one unique finding obtained from this study is a revelation of the scale dependance of chaotic solutions. To determine the relationship (or the association or dependence) of two variables, we calculate the Pearson correlation coefficient (PCC)

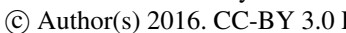

and the Spearman Rank correlation coefficient (SRCC). Both coefficients measure the extent
one variable increases as the other variable increases. The PCC applies an assumption of a
linear relationship, while the SRCC is used as an alternative for examining the dependence
of two variables without an assumption of a strong linear relationship. For PCCs, a positive
high PCC between two variables indicates a strong direct linear relationship, while a very
small PCC suggests a weak "linear" relationship. A PCC of zero indicates no linear relation-
ship. In addition to the PCC and SRCC, scatter plots and linear regression lines are used to
qualitatively display the linear association between two variables. To verify the time lag (or
lead) between two time series (e.g., between the primary mode and tertiary mode), the cross
correlation function (CCF) is calculated. The PCCs (or SRCCs), CCF, and linear regression
lines are calculated using the function "cor" with an option of "pearson" or "spearman",
the function "ccf", and the function "lm", respectively, as provided by R (Adler, 2012). To
minimize the impact of initial model spin up processes, the first 199 time integrations are
not used, giving a starting time of two seconds for the PCC or SRCC calculations.

## 3 Numerical Results

Figure 2 provides the largest ensemble-averaged Lyapunov Exponents (eLEs) as a function
of the forcing parameter $r$ from different LMs. A positive eLE indicates the appearance of
a chaotic solution. The figure displays results obtained from $\triangle r$=1. The pink, black, blue,
and orange lines display the eLEs calculated using the TS method for the 3DLM, 5DLM,
7DLM, and 3DLM-P7d, respectively. The results for the 3DLM and 5DLM are reproduced
from Shen (2014a). As shown in Figure 2, the critical value of $r$ for the onset of chaos is
between r=116 and r=117 for the 7DLM, and between r=149 and r=150 for the 3DLM-P7d.
Additional experiments with a smaller $\triangle r$=0.1 are performed and $r_c$ is found to be 116.9 and
149.2 for the 7DLM and 3DLM-P7d, respectively, as listed in Table 1. These LMs produce
a much larger $r_c$ as compared to the 3DLM and 5DLM. In Figure 2, the yellow open circles
show the eLEs determined by the GSR scheme that display no significant differences from
the eLEs using the TS scheme (as shown with the blue line). Note that the performance of
both the TS and GSR schemes in determining the eLEs of the 3DLM was first documented in
Shen (2014a). For the chaotic regions, the leading eLEs between r=130 and r=160 are close
to 2.0, yielding a Kaplan-York fractal dimension (Kaplan and Yorke, 1979) slightly greater
than 2.20 (not shown). This value is slightly larger than the well-known fractal dimension
of 2.06 for the 3DLM with $r = 28$ and $\sigma = 10$. The solutions from the 3DLM, 5DLM and
7DLM are briefly compared using phase space plots for (Y,Z) in Figure 3. While the 3DLM
with $r = 28$ produces a chaotic solution (Fig. 3a), the 5DLM with ($r = 42$) still yields a
stable solution. With the 7DLM, the solution with $r = 112$ is stable, and the solution with


$r = 120$ is chaotic.

For the 7DLM and 5DLM, we are able to obtain the analytical solutions of critical points
in the phase space, including Eq. (35) for the 7DLM and Eq. (19) of Shen (2014a) for
the 5DLM. Note that the analytical solutions of the 3DLM were first solved in Lorenz
(1963), and were documented in Eq. (21) of Shen (2014a). Here, the analytical solutions
are employed into Eq. (7) to obtain the corresponding solution of the temperature per-
turbation ($\theta$) in the physical space for the 3DLM, 5DLM and 7DLM. Without a loss of
generality, as shown in Figure 4, temperature perturbations are calculated with $r = 120$.
Each of the solutions in different panels is normalized by its maximum, i.e., $max(\theta)$. In all
of the above models, unstable regions (with $\partial\theta/\partial z < 0$) appear near the bottom and top
boundaries (the shaded regions in Figure 4). The unstable layers are thinner in the higher-
dimensional Lorenz models. Near the middle layer, the 3DLM solution is stable, while the
5DLM solution becomes more neutral. The "steepening" of the isotherms is an indication
of stronger convection and can be resolved better with higher wavenumber modes in the
5DLM. In higher-dimensional Lorenz models (i.e., 7DLM), the additional tertiary modes
enable further steepening that leads to the overturning of isotherms in the middle layer
and, thus, produces unstable regions, as shown in the shaded areas in Figure 4. Since the
LMs are forced systems, the advantages of higher-dimensional LMs can clearly be seen in
the better resolved flow patterns in some regions. The differences between these solutions
obtained using different number of modes can be revealed by the PCCs. The PCC of the
analytical solutions for temperature in two different models is calculated for each location
in the horizontal (x) direction using samples in the vertical (z) direction, and, thus, it is a
function of "x". Figure 4d provides PCCs for the temperature perturbations of the 7DLM
and 3DLM (in red), and for the temperature perturbations of the 5DLM and 3DLM (in
black). The two PCC functions which are less than one indicate the differences between
the solutions of the higher-dimensional LM (the 5D or 7D LM) and the baseline LM (the
3DLM). Here, it should be noted that the verification of higher-resolution simulations using
coarser-resolution reanalysis data should be done with caution because small-scale processes
may appear in higher-resolution runs but not in coarser-resolution runs.

The simplicity of the nonlinear analytical solutions may form a good case for testing a
more generalized Lorenz model developed using finite difference or finite volume schemes.
For example, during the initial model development, the analytical solutions for the 3DLM,
5DLM or 7DLM can be used to verify the numerical solutions from the generalized Lorenz
model at comparable resolutions. Using this process, confidence in the performance of the
generalized model can be constructed. Then, a much higher resolution simulation with the
generalized model can be used to better address the question of whether or not a higher
resolution model is more stable or more chaotic. Answering this question will be the topic

of a future study. Next, a linear stability analysis with the analytical solutions is presented.

The above discussions suggest that the 7DLM with $\sigma = 10$ has a much larger $r_c$ for the onset of chaos as compared to the 5DLM and 3DLM. The mathematical analysis provided in Section 2 suggests that the further extension of the nonlinear feedback loop feedback can provide a negative nonlinear feedback to stabilize solutions, consistent with the findings of

Shen (2014a) using the 5DLM. The role of negative nonlinear feedback can also be shown by the 3DLM-P7d that requires a comparable critical value for Rayleigh parameter for the onset of chaos (in Figure 2) as compared to the 7DLM. More detail regarding the 3DLM-P7d analysis is discussed later. Here, it should be noted that the above experiments are performed using a fixed Prandtl parameter ($\sigma = 10$). To examine the impact of negative nonlinear

feedback over the range of Prandtl parameter, a linear stability analysis is performed with respect to the critical point solutions in Eq. (35). Then, eLE analyses, which require significant computational resources, are conducted using the selected values of $\sigma$.

We follow the procedures in the Appendix A of Shen (2014a) to linearize the 7DLM with respect to the critical point solution and construct an eigenvalue problem using the

linear system. Given a pair of $r$ and $\sigma$, the real part of an eigenvalue represents a growth (or decay) rate near the critical point. In N-dimensional Lorenz model, we can obtain N eigenvalues, but only analyze the largest eigenvalue, denoted as $(Re(\lambda))$. Over the range of $r$ and $\sigma$, we can obtain the corresponding largest eigenvalues, and plot the zero contour line for $Re(\lambda)$ in the $(\sigma, r)$ space with a blue line (Figure 5). The values for the $Re(\lambda)$ in

the region above (below) the blue line are positive (negative), suggesting unstable (stable) regions. Therefore, the blue line represents the critical values of $r$ as a function of $\sigma$ for the occurrence of unstable solutions. For example, the intersection of the blue line and the vertical line with $\sigma = 10$ is at $r = 160.3$, suggesting that the linear 7DLM system with $r > 160.3$ may produce an unstable solution. The critical value of "r" determined using the

linear stability analysis is denoted as $r_c^l$, where "$l$" indicates "linear". For each $\sigma$ between 5 and 25, the blue line in Figure 5 suggests that the $r_c^l$ in the 7DLM is always above 50, a value larger than the corresponding $r_c^l$ of the 5DLM indicated by a black line. By comparison, the critical value $(r_c)$ for the onset of chaos determined by the eLE analysis is shown with solid circles. For the selected values of $\sigma$, the corresponding $r_c$ is also above 50, suggesting

a larger $r_c$ for the onset of chaos in the 7DLM than in the 5DLM.

Note that for a $\sigma$ close to 10, the difference between the values of $r_c$ and $r_c^l$ is large. While the $r_c$ is determined by the (nonlinear) eLE analysis (with sufficient number of initial conditions close to the trivial critical point, as shown in Figure A1), the $r_c^l$ is determined using the linear stability analysis close to the non-trivial critical point. The difference in $r_c$

and $r_c^l$ is further illustrated by performing the following two sets of experiments. The first set of experiments are initialized at the trivial point with initial perturbations of (Y, Z) as (1,-1)



(i.e., Eq. 36a). The $r_c$ determined by the first set of experiments is 116.9 (e.g., Figure 6) and is compared well with the eLE analysis. The second set of experiments are initialized at the positive non-trivial critical point (Eq. 35) using the same initial perturbations of (Y,Z) as

(1,-1) (i.e., Eq. 36b). The $r_c$ from the second set of experiments is 160.8 (e.g., Figure 7) and is comparable to that obtained using the linear stability analysis. Note that Figure 7 shows the results from a long time integration with $\tau = 1800$. Based on the above discussions, the difference between $r_c$ and $r_c^l$ may indicate the dependence of $r_c$ on the initial position (initial conditions), and may indicate a deficiency in the calculation of solution stability using finite-

time integration. However, when the 10,000 ensemble of initial conditions are multiplied by a factor of 100, and the eLE calculations are performed, the results yield a comparable $r_c$ to that obtained using the original initial conditions (as shown in Figure A2). Therefore, differences between the $r_c$ and $r_c^l$ for $\sigma \sim 10$ deserve further examination. Practically, when $r_c^l$ is much larger than $r_c$, it takes a much longer time period for the numerical solution

to become unstable when the corresponding simulation is initialized near the non-trivial critical point instead of the trivial critical point. In other words, a simulation initialized at the non-trivial critical point has a larger predictability than one initialized at the trivial critical point.

Previously, we suggested that the negative nonlinear feedback, which can help stabilize solutions, can be emulated using the parameterized term $Q_2$ in Eq. (32) of the revised 3DLM. To obtain this term, we made the assumption of the quasi-equilibrium state solutions for the secondary and tertiary modes, leading to Eqs. (25-26) and Eqs. (27-28), respectively. These equations can help illustrate the relationship between the primary, secondary and tertiary modes when higher wavenumber modes provide feedback. With further assumptions of $X^2 >> bd_o$ and $X^2 >> bd_1$ in Eqs. 32, 26 and 28, we obtain:

$$Q_2 \approx -2bZ, \tag{37}$$

$$Z_2 \approx \frac{2Z_1}{3}, \tag{38}$$

and

$$Z_1 \approx \frac{Z}{2}. \tag{39}$$

Therefore, conceptually, when a negative nonlinear feedback is emulated by Eq. (37) in

a special revised 3DLM, the system displays a strong positive linear relationship between $Z_2$ and $Z_1$ and between $Z_1$ and $Z$. In Eq. (38) (or Eq. 39) the corresponding PCC is equal to one, as the PCC measures the linear relationship of the two variables with any non-zero slope. Note that the scale dependence in the special revised 3DLM is implicit because the secondary and tertiary modes are not explicitly included. In addition, the

assumptions of quasi equilibrium (i.e., $\partial/\partial\tau \sim 0$) for the secondary and tertiary modes may pose a challenge in representing chaotic responses (i.e., rapid changes in time) using these

higher wavenumber modes. In spite of the deficiency, the above analysis qualitatively reveals interesting features of scale dependence. Next, we examine the scale dependence for chaotic solutions in the 7DLM.

As discussed previously, the 7DLM with $r = 120$ produces chaotic solutions that, as shown by the trajectories of solutions in the phase space, result in a butterfly pattern. This pattern is also shown in scatter plots between the two variables (e.q., $Y_1$ and $Z_1$ or $Y_2$ and $Z_2$) in Figure 8. The corresponding PCCs provided in the bottom-right of each panel are very small, indicating no linear relationship. The regression line with a nearly zero slope, as

shown by a red line in Figure 8, also suggests no linear association.

In general, since the 7DLM is a nonlinear model, the secondary (tertiary) temperature mode Z1 (or Z2) is assumed to be a nonlinear function of the primary temperature mode ($Z$). Obtaining a small correlation between two variables for the transient (chaotic) solutions is not out of the ordinary. Interestingly, in the following discussion, a scale dependence

with a strong linear relationship between two modes in the 7DLM and the 5DLM will be presented.   Figure 9 provides scatter plots for $Y$ vs. $Y_1$ and $Z$ vs. $Z_1$ from the 5DLM and 7DLM with $r = 120$, producing chaotic solutions. For the 5DLM, the PCC of $Y$ and $Y_1$ ($Z$ and $Z_1$) is 0.803 (0.954). Results with high PCCs suggest that the primary and secondary temperature modes have a strong direct linear relationship (Figures 9a-9b) and,

thus, indicate a scale dependance. Within the 7DLM that includes the tertiary temperature modes, the corresponding PCCs between the primary and secondary temperature modes (e.g., $Y$ vs. $Y_1$ and $Z$ vs. $Z_1$) become larger, suggesting a stronger linear relationship (e.g., Figures 9c-d). The strong relationship among different scale modes is also shown in the SRCCs listed in Table 2.

Since the 7DLM includes three major scales containing the primary, secondary, and tertiary temperature modes, the scale dependance (or correlation) among them is further analyzed in Figure 10. The PCCs between the secondary and tertiary modes, which has a value of 0.967 for $Y_1$ and $Y_2$ and a value of 0.998 for $Z_1$ and $Z_2$, respectively, are higher (Figures 10a-b) than the PCCs between the primary and secondary modes (Figures 9c-d). The PCC

between the primary and secondary modes is larger than the PCC between the primary and tertiary modes (in Figures 10c-d). These results indicate a hierarchical scale dependence. Scale dependence is also clearly observed in the cross correlation function between $Z$ and $Z_1$; and in the time evolution of $Z$, $Z_1$ and $Z_2$ in Figure 11 where a much longer integration time ($\tau = 1000$) is used. Table 2 provides PCCs and SRCCs from additional experiments with

different Rayleigh numbers (e.g., $r = 140$ and $r = 160$) that produce chaotic solutions. The results indicate similar scale dependence. As discussed in Shen (2014a) and in this study, the secondary modes $Y_1$ and $Z_1$ are added into the 5DLM in order to extend the nonlinear feedback loop of the 3DLM. The tertiary modes $Y_2$ and $Z_2$ are included in the 7DLM to



further extend the nonlinear feedback loop of the 5DLM. Therefore, the occurrence of the hi-
erarchical scale dependence is indeed related to the extension of the nonlinear feedback loop
in the 7DLM. Therefore, a higher-dimensional Lorenz model (e.g., 9DLM) with a further
extension of the nonlinear feedback loop may display a stronger scale dependence.

## 4 Conclusions

With this study, the impact of an extended nonlinear feedback loop is discussed and hier-
archical scale dependence using the 7D Lorenz model is revealed. Based on the analysis of
the nonlinear Jacobian term, the 7DLM is constructed to include seven Fourier modes that
possess three major scales, including primary temperature modes (i.e., $Y$ and $Z$), secondary
temperature modes (i.e., $Y_1$ and $Z_1$) and tertiary temperature modes (i.e., $Y_2$ and $Z_2$). As
the high wavenumber modes are selected to extend the nonlinear feedback loop in the 3DLM
and 5DLM, the 7DLM could be reduced to be the lower-dimensional Lorenz models when the
higher wavenumber modes are neglected. Previously, Shen (2014a) demonstrated that the
extension of the nonlinear feedback loop in the 5DLM can provide negative nonlinear feed-
back to stabilize solutions. In this study, we illustrate a similar role of the negative nonlinear
feedback in stabilizing the solutions in the 7DLM. The critical value of the Rayleigh param-
eter ($r_c = 116.9$) for the onset of chaos in the 7DLM is large as compared to the $r_c = 24.74$
of the 3DLM and the $r_c = 42.9$ of the 5DLM. The impact of negative nonlinear feedback
is further illustrated using the revised 3DLM with a parameterized term that emulates the
negative feedback. The parameterized term is obtained by assuming quasi equilibrium state
solutions for the secondary and tertiary modes and expressing these modes in terms of the
primary modes. The results indicate that the $r_c$ of the revised 3DLM is comparable to that
of the 7DLM and much larger than that of the 3DLM. We are able to solve for the analyt-
ical solutions of critical points in both 5DLM and 7DLM. By linearizing these two models
with respect to the critical points and constructing eigenvalue problems, we perform linear
stability to show that the 7DLM requires a larger Rayleigh parameter for onset of chaos
when $5 \leq \sigma \leq 25$. The eLE analysis with selected values of $\sigma$ yields the same conclusion.

While the 7DLM produces a chaotic solution with a relatively large r (e.g., r=120), a
hierarchical scale dependence appears in the solutions. Such a result is indicated by ele-
vated correlation coefficients between the primary and secondary modes (i.e., $Z$ and $Z_1$)
and between the secondary and tertiary modes (i.e., $Z_1$ and $Z_2$), with the latter larger than
the former. For example, for chaotic solutions with r=120, the Pearson correlation coeffi-
cients (PCCs) between the primary and secondary modes (i.e., Z and Z1) and between the
secondary and tertiary modes (i.e., Z1 and Z2) are 0.988 and 0.998, respectively. Note that
the high correlations between $Z$, $Z_1$ and $Z_2$ do not suggest causality, and, therefore, do not



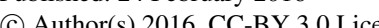



suggest a control and feedback relationship. However, the results may suggest that dissipa-

tive processes associated with $-bZ$, $-4bZ_1$ and $-9bZ_2$ are coherent (in phase in time). The scale dependence is consistent with Eqs. (25-28) that are obtained under the assumption of quasi equilibrium for the secondary and tertiary modes, and used to emulate the negative nonlinear feedback (Eq. 32).

In conclusion, the nonlinear feedback loop can be extended using the new modes that

enable the successive downscale and upscale transfer described by the Jacobian term. The 7DLM, constructed by extending the nonlinear feedback loop of the 5DLM, reveals the role of the associated negative nonlinear feedback in stabilizing solutions and the hierarchical scale dependence in chaotic solutions. Future work will examine how higher dimensional LMs may produce larger critical values for the Rayleigh parameter for the onset of chaos

while displaying a stronger hierarchical scale dependence.

*Acknowledgements.* I thank X. Zeng, R. Pielke Sr., and C. Interlando for valuable comments and encouragement, and Ms. E. K. Yoo for her help in the derivations and verification of the seven-dimensional Lorenz model. I appreciate R. Carretero's kindness in inviting me to give a lecture regarding high-order Lorenz models and providing valuable comments during spring 2015. I am

grateful for support from the College of Science at San Diego State University and the NASA Advanced Information System Technology (AIST) Program. Resources supporting this work were provided by the NASA High-End Computing (HEC) program and the NASA Advanced Supercomputing division at Ames Research Center.

**Appendix A: Initial conditions for the eLE calculation**

In this section, we discuss how the 10 000 initial conditions are generated for calculation of the ensemble Lyapunov exponents and their impact on determining the critical value of Rayleigh number for the onset of chaos. The 10 000 different ICs are produced as Gaussian white noise with the center at the trivial critical point (i.e., with a mean value of zero for the ICs). The method is described by Press et al. (1992) and the Fortran code was kindly

provided by Professor Z. Wu of Florida State University. Figure A1 shows the 10 1000 ICs. For the 3D, 5D, or 7D LM with a given r and the 10 0000 ICs, it takes approximately 10-20 wall-time hours to obtain an eLE. To efficiently calculate eLEs over a wide range of r, a simple task-level parallelism is implemented in order to perform parallel calculations using multiple computing processors on the National Aeronautics and Space Administration

(NASA) supercomputers (e.g., Biswas et al. 2007; Shen 2014a).

In Figure 2, we discussed how the critical value of Rayleigh number ($r_c$) is determined using the analysis of the eLEs that are calculated using the 10 000 ICs shown in Figure A1. As discussed in section 2, a large number of ICs is used to minimize the impact of



ICs on the eLE calculation and, thus, the estimate of $r_c$. The run with the 7DLM using

the ICs is used as the control run for a further comparison. Since the eLE is calculated
using finite time integrations, parallel experiments are performed with the 7DLM and the
results are compared to the control run in order to further examine the impact of ICs on
the eLE calculation. For the parallel experiments, the 10 000 ICs of the control run are
multiplied by 100, which allows ICs to be distributed over a larger space. As shown with

green and orange lines in Figure A2, eLE calculations are performed, respectively, using the
TS and GSC methods. The eLE calculations performed using the two methods produce
comparable results with minimal differences. A comparison between the parallel runs and
control run shows differences in the eLE calculations over $100 \leq r \leq 150$, indicating the
impact of ICs. However, the $r_c$ determined using the parallel experiments is between 113

and 114, only slightly smaller than the $r_c$ ($\sim 116.9$) obtained from the control run. These
parallel experiments provide additional support for the determination of $r_c$ using the eLE
analysis in the 7DLM.





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





**Table 1.** The characteristics of various Lorenz models. The "Equations" column provides a list of the equations used in each specific Lorenz model. Values for $r_c$ and $r_c^l$ are determined based on the eLE analyses and the linear stability analysis, respectively. The "Scaling factors" column indicates the analytical solitons of the critical points used to normalize the solutions in Figure 3. * For the 3DLM, the ensemble averaged LE is $1.2 \times 10^{-2}$ at $r = 23.7$, and becomes 0.26 at $r = 24$.

| Model | Equations | Figures | $r_c$ | $r_c^l$ | Scaling factors |
|---|---|---|---|---|---|
| 3DLM | Eqs. (15)–(17) of Shen (2014a) | 2–4 | 23.7* | 24.74 | Eq. (21) of Shen (2014a) |
| 3DLMP5d | Eqs. (1)–(3),(10) of Shen (2015a) | | 52.1 | N/A | Eq. (19) of Shen (2014a) |
| 3DLMP7d | Eqs. (29)–(32) | 2 | 149.2 | N/A | Eqs. (35a)–(35g) |
| 5DLM | Eqs. (10)–(14) of Shen (2014a) | 2–5, 9 | 42.9 | 45.94 | Eq. (19) of Shen (2014a) |
| 7DLM | Eqs. (11)–(17) | 2–11 | 116.9 | 160.3 | Eqs. (35a)–(35g) |

**Table 2.** The Pearson correlation coefficients (PCC) and the Spearman Rank correlation coefficient (SRCC) between two different variables from the primary, secondary, and tertiary modes in the 5DLM or 7DLM. For the PCCs, results obtained from three cases with $r = 120$, 140 and 160 are provided. For the SRCCs, only results from the case with $r = 120$ are listed.

| Model | Variables | PCC r=120 | SRCC r=120 | PCC r=140 | PCC r=160 |
|---|---|---|---|---|---|
| 5DLM | $Y - Y_1$ | 0.803 | 0.751 | 0.796 | 0.785 |
| Same | $Z - Z_1$ | 0.954 | 0.944 | 0.952 | 0.955 |
| 7DLM | $Y - Y_1$ | 0.861 | 0.771 | 0.861 | 0.865 |
| Same | $Y - Y_2$ | 0.731 | 0.684 | 0.724 | 0.725 |
| Same | $Y_1 - Y_2$ | 0.967 | 0.971 | 0.963 | 0.962 |
| Same | $Z - Z_1$ | 0.988 | 0.975 | 0.986 | 0.985 |
| Same | $Z - Z_2$ | 0.984 | 0.965 | 0.981 | 0.987 |
| Same | $Z_1 - Z_2$ | 0.998 | 0.998 | 0.998 | 0.998 |





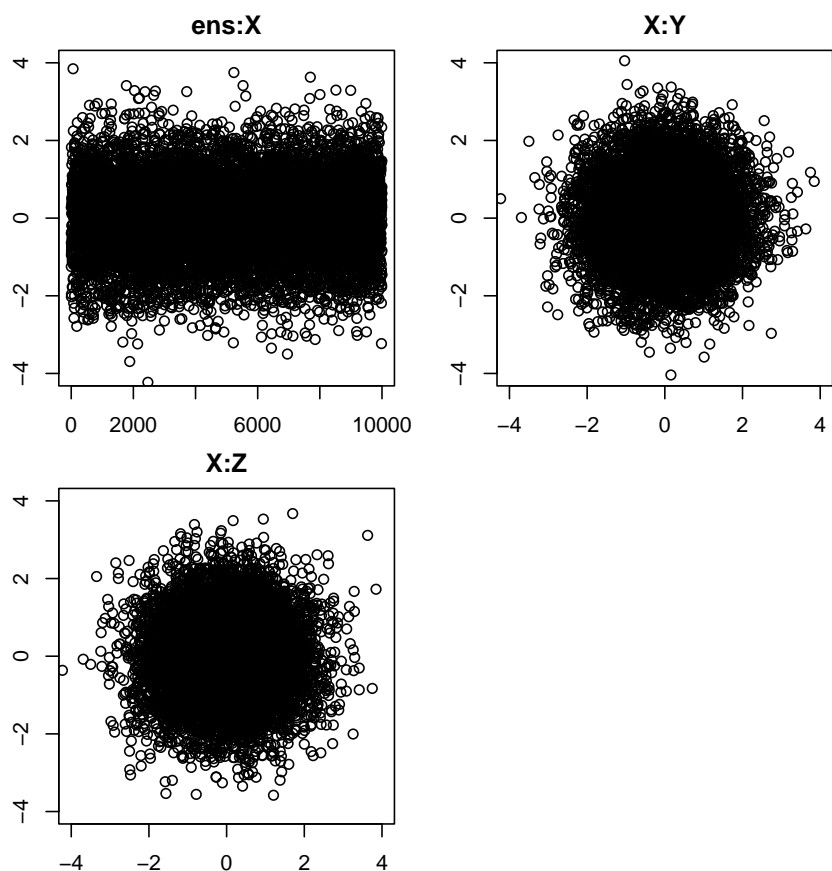

Figure A1: 10,000 initial conditions for calculation of the ensemble Lyapunov exponent. (a) The distribution of X as a function of the ensemble members. (b) The distributions of X and Y. (c) The distributions of X and Z.





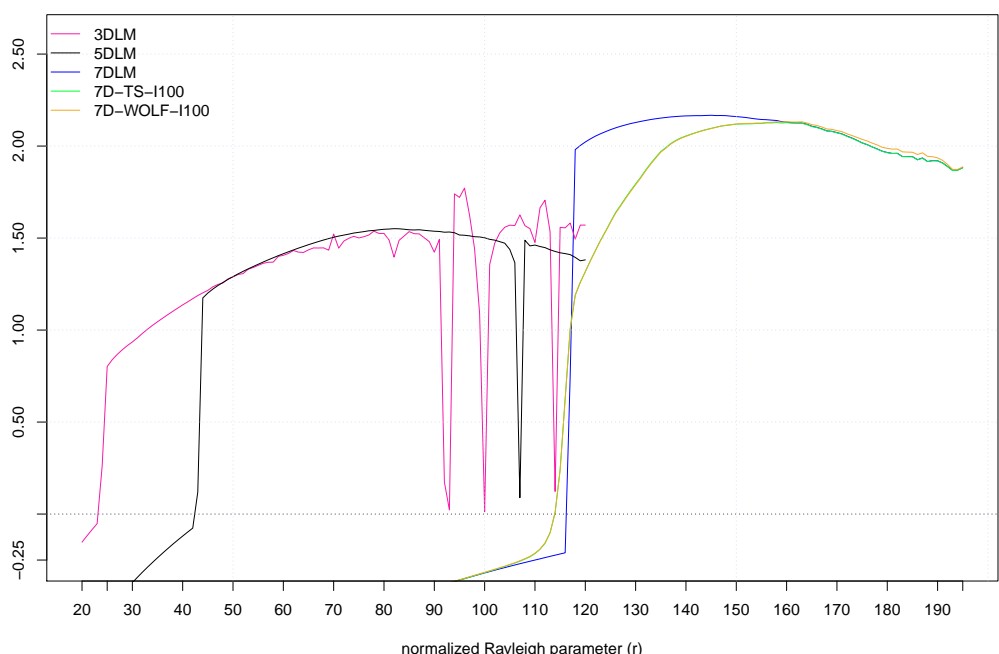

Figure A2: The largest ensemble-averaged Lyapunov Exponents (eLEs) as a function of the forcing parameter, $r$, in different LMs. The figure provides results for $\triangle r = 1$. The pink, black, and blue lines display eLEs for the 3DLM, 5DLM, and 7DLM, respectively, using the TS method. Two parallel experiments using the 7DLM with ICs that are 100 times that of the ICs in the control run are shown with green and orange lines. The first one applies to the TS method, while the second uses the GSR method. Note that the critical value of $r$ for the onset of chaos in the 7DLM with different ICs are comparable.





Figure 1: A schematic diagram of an extended feedback loop which consists of the downscaling and upscaling processes associated with $J(M_1, M_j)$, $j = 2, 3, 5, 6, 8$ or, 9. For a given $M_j$ mode, $J(M_1, M_j)$ may lead to a downscaling process, as indicated by a downward arrow, and/or an upscaling process, as indicated by an upward arrow. The nonlinear feedback loop in the 3DLM and its extension within the 5DLM are shown with pink and blue arrows, respectively (e.g., Shen 2014a). Further extension of the nonlinear feedback loop within the 7DLM is shown with orange arrows. A number in parentheses is the coefficient of the specific mode. The "$M_3(ml)$" in the leftmost column represents that the $M_3$ mode with a coefficient of "ml" is generated or influenced by a downscaling process from $J(M_1, M_2)$. The terms $-XZ$, $-XY_1$, and $-2XY_2$, which appear in Eqs. (12), (13), and (15), are associated with the upscaling process of the $J(M_1, M_3)$, $J(M_1, M_5)$, and $J(M_1, M_8)$ that are indicated by the pink, blue, and orange upward arrows, respectively. $\otimes^{3D}$, $\otimes^{5D}$ and $\otimes^{7D}$ indicate the end of downscaling due to mode truncation in the 3DLM, 5DLM, and 7DLM, respectively.





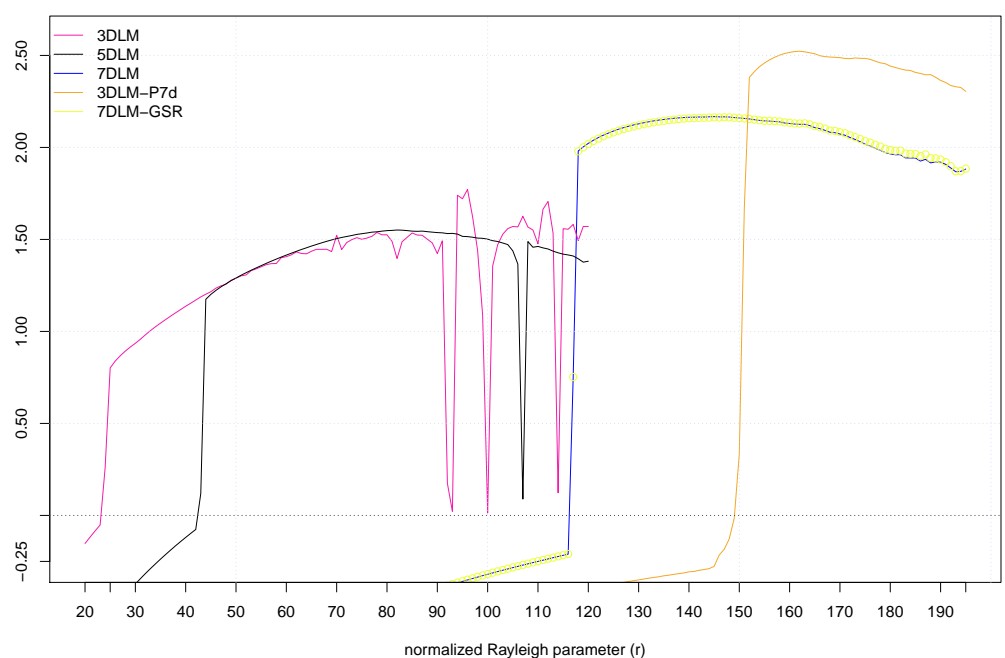

Figure 2: The largest ensemble-averaged Lyapunov Exponents (eLEs) as a function of the forcing parameter, $r$, in different LMs. The figure provides results for $\triangle r$=1. The pink, black, blue, and orange lines display eLEs for the 3DLM, 5DLM, 7DLM, and 3DLM-P7d, respectively, using the TS method. Open yellow circles indicates the eLEs of the 7DLM using the GSR scheme. For the 7DLM, both methods yielded comparable results. The appearance of chaotic solutions is indicated by positive eLEs. Note that the critical value of $r$ for the onset of chaos in the 7DLM is between r=116 and r=117. The pink and black lines are reproduced from Shen (2014a).





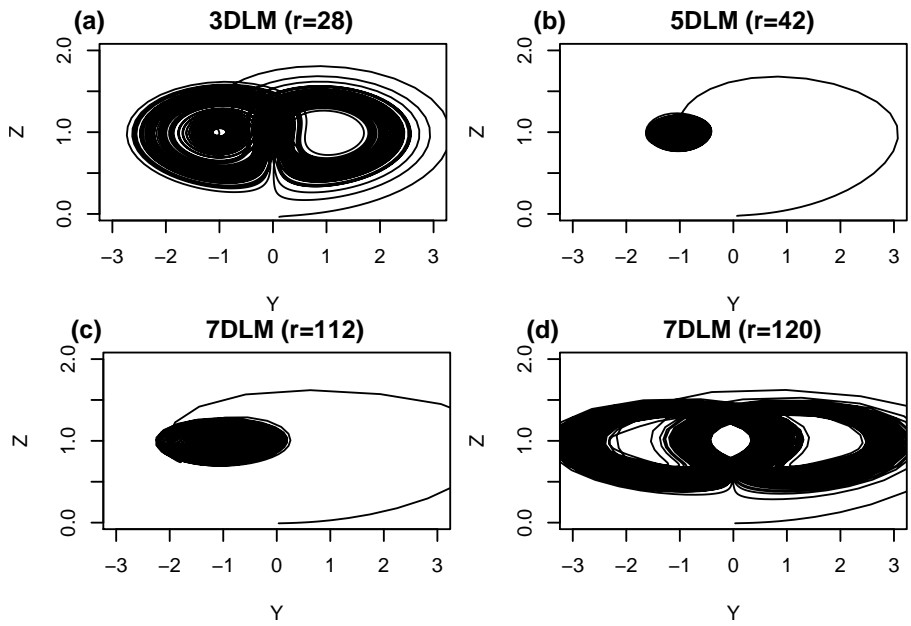

Figure 3: Phase space plots for $(Y, Z)$ in the different Lorenz models. (a) Lorenz strange attractors with r=28 in the 3DLM. (b) A stable solution with r=42 in the 5DLM. (c-d) Stable and chaotic solutions with r=112 and r=120 in the 7DLM, respectively. All of the solutions are normalized by the the corresponding critical points, namely, Eq. (21) of Shen (2014a) for the 3DLM, Eq. (19) of Shen (2014a) for the 5DLM, and Eq. (35) for the 7DLM.





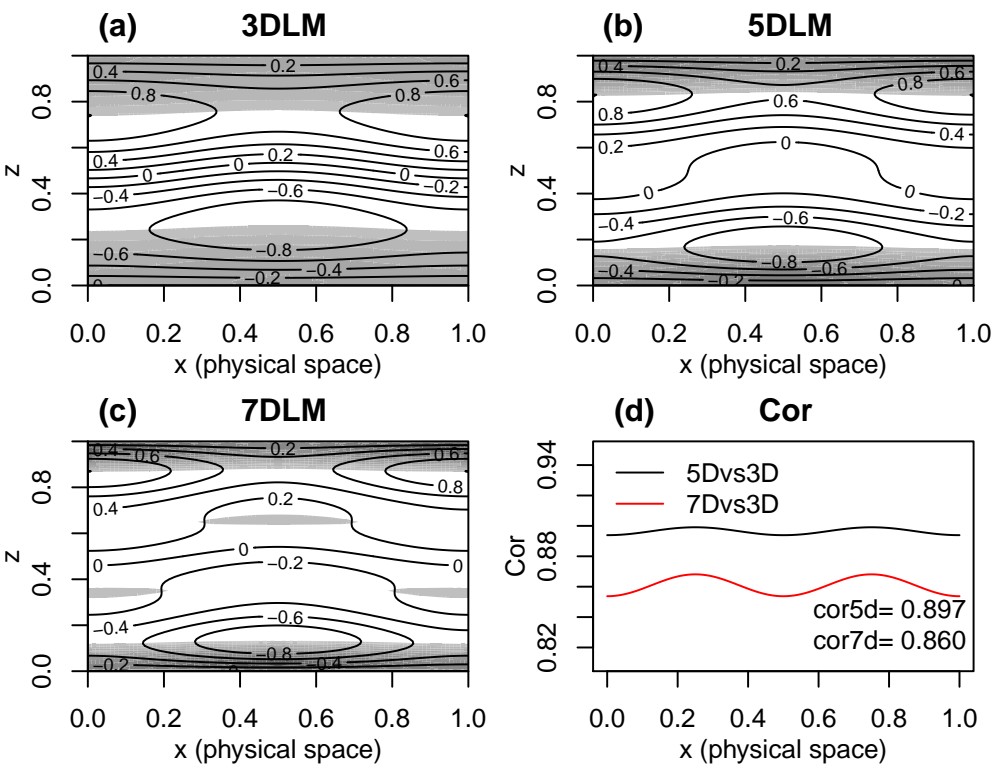

Figure 4: Temperature perturbation in the spatial space $(x, z)$, represented by the analytical solutions of critical points in the 3DLM, 5DLM, and 7DLM (a-c). (d) Correlation coefficients, calculated for each x location using samples in the z direction.





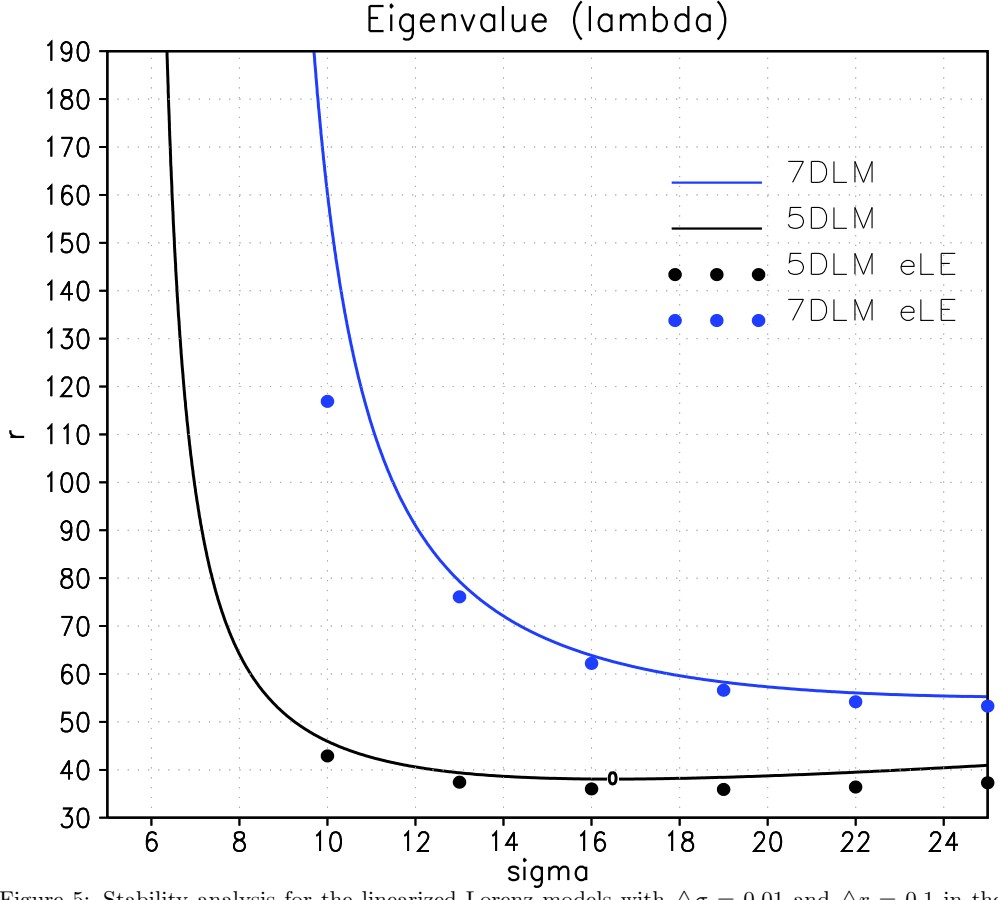

Figure 5: Stability analysis for the linearized Lorenz models with $\triangle\sigma = 0.01$ and $\triangle r = 0.1$ in the 5DLM and 7DLM. The leading eigenvalue $\mathrm{Re}(\lambda)$ as a function of $\sigma$ and $r$. The black and blue lines indicate a constant contour of $\mathrm{Re}(\lambda) = 0$ for the linear 5DLM and 7DLM, respectively. Solid circles with the same color scheme indicate a $r_c$ determined by the eLE analysis with $\triangle r = 1.0$ in the corresponding nonlinear LM.




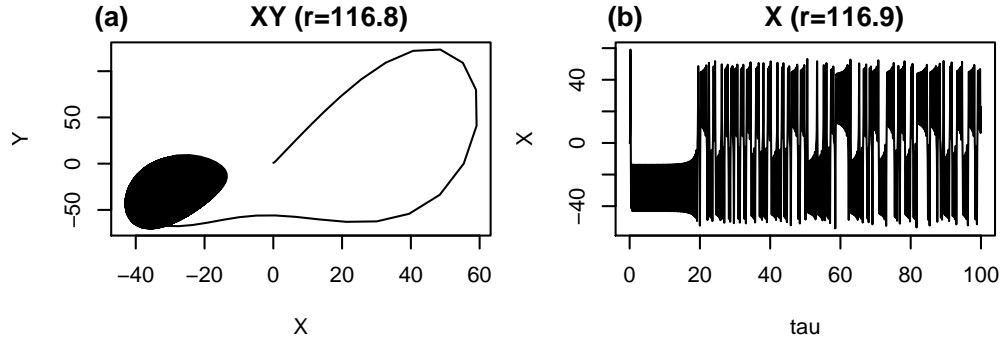

Figure 6: A transition from a stable solution with $r = 116.8$ to a chaotic solution with $r = 116.9$ in the 7DLM. The initial conditions are given in Eq. (36a) (i.e., $(X, Y, Z, X_1, Y_1, Z_1, Y_2, Z_2) = (0, 1, -1, 0, 0, 0, 0, 0)$).





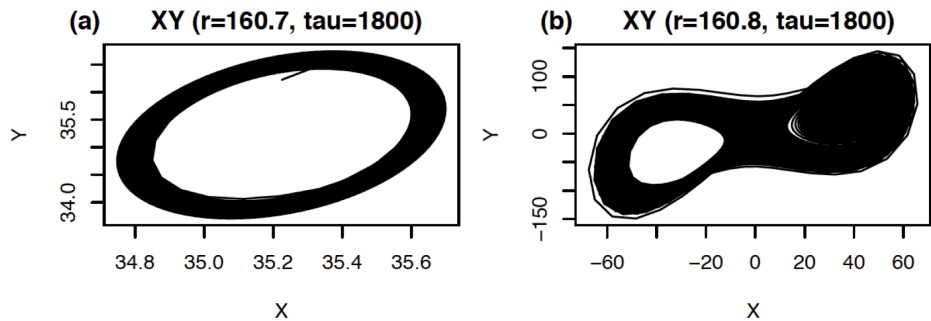

Figure 7: A transition from a stable solution with $r = 160.7$ to a chaotic solution with $r = 160.8$ in the 7DLM. The initial conditions are given near the critical point in Eq. (36b) (i.e., $(X, Y, Z, X_1, Y_1, Z_1, Y_2, Z_2) = (X_c, Y_c+1, Z_c-1, Y_{1c}, Z_{1c}, Y_{2c}, Z_{2c})$). Note that a much larger integration time is used in the simulations.





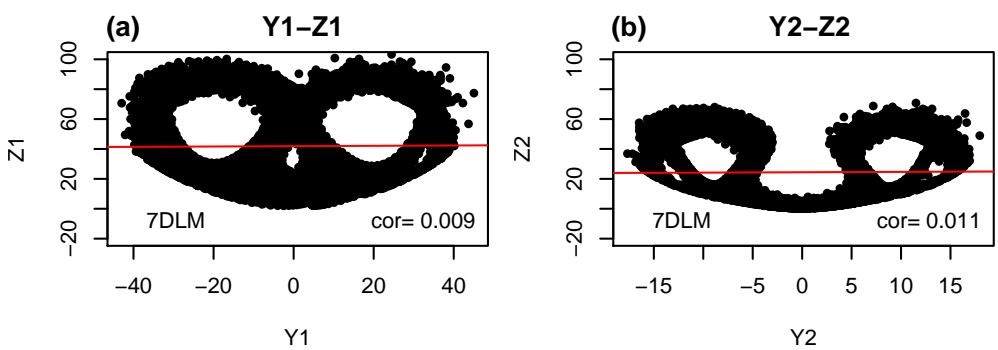

Figure 8: Scatter plots for $Y_1$ vs. $Z_1$ (a) and $Y_2$ vs. $Z_2$ (b) from the 7DLM with $r = 120$.





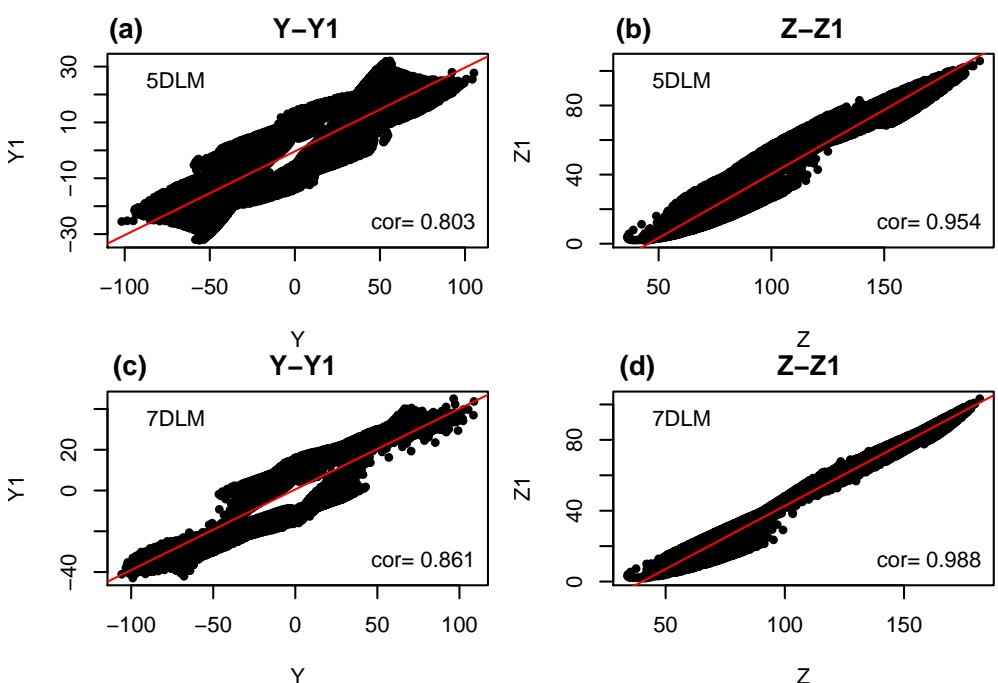

Figure 9: Scatter plots for the 5DLM and 7DLM with $r = 120$. The results indicate that the correlation coefficients become larger in the 7DLM than in the 5DLM.





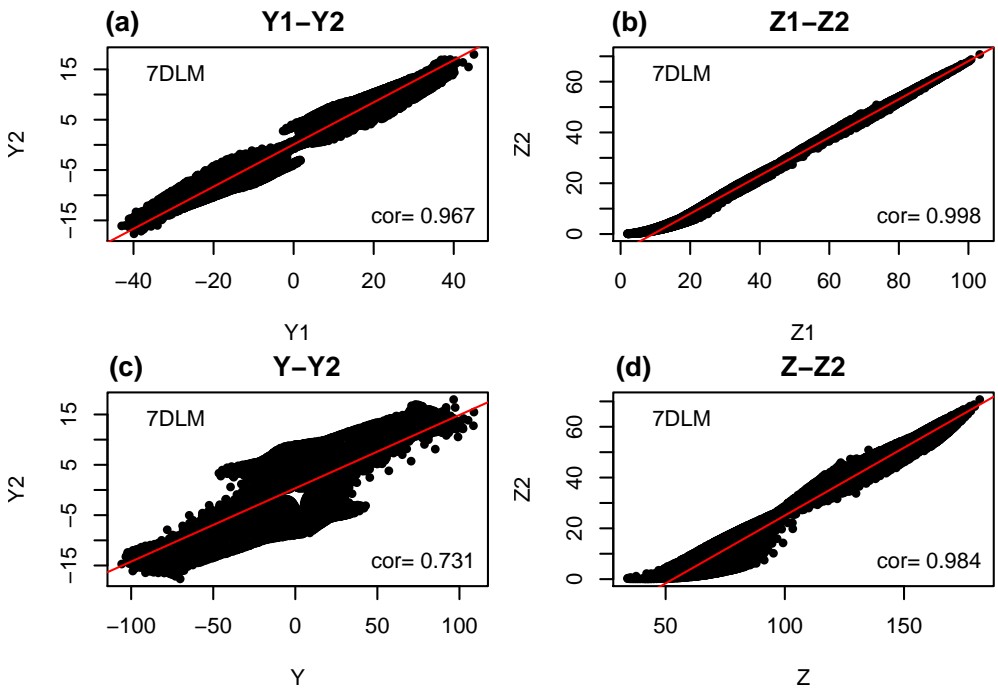

Figure 10: Scatter plots for the 7DLM with $r = 120$. The correlation coefficient for the secondary and tertiary modes (i.e., $Y_1$ and $Y_2$ or $Z_1$ and $Z_2$ in the top panels) is larger than that of the primary and secondary modes (i.e., $Y$ and $Y_1$ or $Z$ and $Z_1$ in the bottom panels of Figure 9).




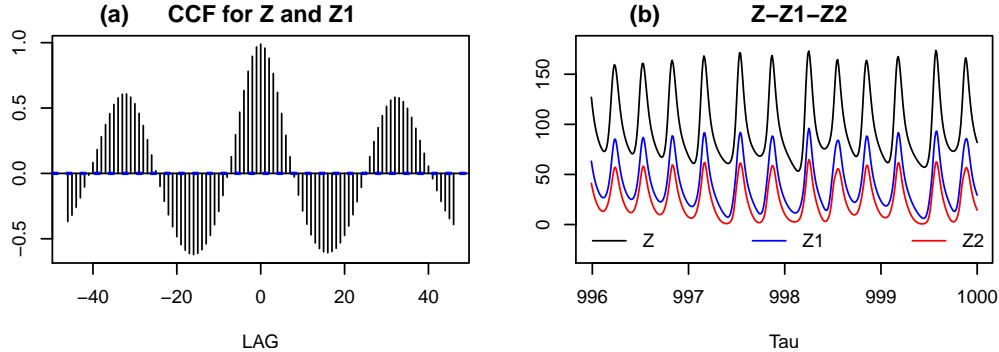

Figure 11: The cross correlation function (CCF) for two time series ($Z$ and $Z_1$) (a) and the time evolution of $Z$, $Z_1$, and $Z_2$ (b) from the 7DLM with r=120. The total integration time is $\tau = 1000$.