# Peer review of "Hierarchical scale dependence associated with the extension of the nonlinear feedback loop in a seven-dimensional Lorenz model"

_Nonlinear Processes in Geophysics, 2016_

## Short Comment (SC1) · 18 May 2016

This is a very interesting and well written paper in which the author extends his previous work on the generalized (5D and 6D) Lorenz models to a 7D model. As compared to the original 3D Lorenz model, 4 extra high wavenumber modes are added at three different major scales: the largest (primary), middle (secondary) and smallest (tertiary). The author performs extensive studies of the effects caused by these additional modes on the 7D system, with a special emphasis on negative nonlinear feedback and the smallest scale modes, and their role in stabilizing the solutions. As a result of this stabilization, the onset of chaos occurs at much higher values of the Rayleigh parameter than for the 3D and 5D systems. This increase of stability of solutions with the
number of added modes is a known effect, nevertheless, the paper gives a new significant insight into the role played by different added modes in stabilizing the solutions. Moreover, a hierarchical scale dependence that appears in the solutions is investigated in details by computing the Pearson correlation coefficients between the primary and secondary, and between the secondary and tertiary modes.

As possible author's future work, I'd like to suggest to explain specifically the fact that the Rayleigh parameter for the onset of chaos in the 7D system increases by a factor of 3 (approximately) as compared to the 5D system, and by a factor of 5 (approximately) when compared to the original 3D Lorenz system. Moreover, I'd like to see discussion of changes of routes to chaos in these higher dimensional Lorenz systems as compared to the original Lorenz system, and what is the physics behind such changes.

In summary, I find this paper an important contribution to studies of the generalized Lorenz systems and to our understanding of the origin of chaos in these systems, and possible ways to control it. I do hope that the paper is published in NPGD very soon!

Zdzislaw Musielak Professor of Physics University of Texas at Arlington

NPGD

---

## Referee Comment (RC1) · Anonymous Referee #1 · 25 May 2016

This is a nice study. By extending the Lorenz systems to seven-dimension versions that include the different-scale modes, author analyzed the impacts of newly resolved small processes and the additional nonlinear terms on the stability of nonlinear Lorenz system in details. He emphasized the negative nonlinear feedback of small-scale temperature perturbation to stabilize solution. Manuscript is well organized and statements are clear and precise. This is an interesting and useful contribution to NPG and my recommendation is to accept.

Pg.19, Line 555: is ->are
* * *
[Figure]

2016.

---

## Referee Comment (RC2) · Y. WU (Referee) · 4 Jun 2016

In this paper the author extends the 5d Lorenz model to 7d Lorenz model by adding two more high wavenumber modes to the temperature perturbation. The addition of such small-scale perturbations in temperature provides negative nonlinear feedback to stabilize solutions is emphasized. Moreover, the hierarchical scale dependence of chaotic solutions between the pure vertical terms in temperature perturbation is revealed. Both results are helpful to our understanding of the chaos phenomenon. The paper is well organized and good written and I recommend it for publication.

Small comments: 1. On page 3 Ln 43 the author explains the butterfly effect of the first kind, and mentions the butterfly effect of the second kind without explanation on Ln 45.

At least a reference should be given here. 2. On Page 3 Ln 58 the author mentioned the data assimilation schemes have been developed to improve the initial conditions. Actually the data assimilation schemes can also be used to optimize the parameters in a dynamical system. 3. To which extent the approximation of equation (9) stands? Any references? 4. On page 16 the lines are not correctly labeled between 440 and 445.

---

## Author Comment (AC1)

Responses to Reviewer I's comments:

General comments:

> This is a nice study. By extending the Lorenz systems to seven-dimension versions that include the different-scale modes, author analyzed the impacts of newly resolved small processes and the additional nonlinear terms on the stability of nonlinear Lorenz system in details. He emphasized the negative nonlinear feedback of small-scale temperature perturbation to stabilize solution. Manuscript is well organized and statements are clear and precise. This is an interesting and useful contribution to NPG and my recommendation is to accept.

Thanks for your comments very much.

> Pg.19, Line 555: is ->are

I checked this part again and believe that it is correct. Thanks.

---

## Author Comment (AC2)

Responses to Reviewer II's comments:

General comments:

> In this paper the author extends the 5d Lorenz model to 7d Lorenz model by adding two more high wavenumber modes to the temperature perturbation. The addition of such small-scale perturbations in temperature provides negative nonlinear feedback to stabilize solutions is emphasized. Moreover, the hierarchical scale dependence of chaotic solutions between the pure vertical terms in temperature perturbation is revealed. Both results are helpful to our understanding of the chaos phenomenon. The paper is well organized and good written and I recommend it for publication.

Thanks for your comments.

Specific comments:

> 1. On page 3 Ln 43 the author explains the butterfly effect of the first kind, and mentions the butterfly effect of the second kind without explanation on Ln 45. C1 NPGD Interactive comment Printer-friendly version Discussion paper At least a reference should be given here.

Thanks for your help. The related sentences have been revised as follows:

*In our previous studies, this feature is referred to as the butterfly effect of the first kind (e.g., Shen 2014a, 2015b). Based on the numerical phenomenon, it has been inferred that tiny perturbation may alter the large-scale flow (e.g., producing a tornado in Lorenz, 1972), which is referred to as the butterfly effect of the second kind (e.g., Pielke 2008; Shen 2014a).*

> 2. On Page 3 Ln 58 the author mentioned the data assimilation schemes have been developed to improve the initial conditions. Actually the data assimilation schemes can also be used to optimize the parameters in a dynamical system.

The related sentences have been revised as follows:

*To minimize the negative impact of inaccurate initial conditions and to optimize parameters in a dynamical system,*

3. To which extent the approximation of equation (9) stands? Any references?

Thanks for your help. The derivation of Eq. (9) is provided below. To be more precise, I replace "≈" by "=" in Eq. (9) of the revised manuscript. Results remain unchanged.

$$
\begin{aligned}
J(M_1, M_6) &= \left( \frac{\partial M_1}{\partial x} \frac{\partial M_6}{\partial z} - \frac{\partial M_1}{\partial z} \frac{\partial M_6}{\partial x} \right) \\
&= \frac{\partial M_1}{\partial x} \frac{\partial M_6}{\partial z} \\
&= 4\sqrt{2} ml \cos(lx) \sin(mz) \cos(4mz) = 2\sqrt{2} ml \cos(lx) \Big( \sin(5mz) - \sin(3mz) \Big) \\
&= 2ml \Big( \sqrt{2} \cos(lx) \sin(5mz) - \sqrt{2} \cos(lx) \sin(3mz) \Big) \\
&= 2ml(M_8 - M_5)
\end{aligned}
$$

4. On page 16 the lines are not correctly labeled between 440 and 445.

Thanks for your help. This has been fixed in the revised manuscript, which has been submitted.

---

## Author Comment (AC3)

Responses to short comments:

General comments:

> This is a very interesting and well written paper in which the author extends his previous work on the generalized (5D and 6D) Lorenz models to a 7D model. As compared to the original 3D Lorenz model, 4 extra high wavenumber modes are added at three different major scales: the largest (primary), middle (secondary) and smallest (tertiary). The author performs extensive studies of the effects caused by these additional modes on the 7D system, with a special emphasis on negative nonlinear feedback and the smallest scale modes, and their role in stabilizing the solutions. As a result of this stabilization, the onset of chaos occurs at much higher values of the Rayleigh parameter than for the 3D and 5D systems. This increase of stability of solutions with the number of added modes is a known effect, nevertheless, the paper gives a new significant insight into the role played by different added modes in stabilizing the solutions. Moreover, a hierarchical scale dependence that appears in the solutions is investigated in details by computing the Pearson correlation coefficients between the primary and secondary, and between the secondary and tertiary modes.

Thanks for your comments.

Specific comments:

> As possible author's future work, I'd like to suggest to explain specifically the fact that the Rayleigh parameter for the onset of chaos in the 7D system increases by a factor of 3 (approximately) as compared to the 5D system, and by a factor of 5 (approximately) when compared to the original 3D Lorenz system. Moreover, I'd like to see discussion of changes of routes to chaos in these higher dimensional Lorenz systems as compared to the original Lorenz system, and what is the physics behind such changes.

Thanks for your comments and suggestions very much. I will address these using the 7DLM as well as two 9D LMs in a future study, since I focused on the role of the negative nonlinear feedback in stabilizing solution and the appearance of the hierarchical scale dependence in the current study.

In summary, I find this paper an important contribution to studies of the generalized Lorenz systems and to our understanding of the origin of chaos in these systems, and possible ways to control it. I do hope that the paper is published in NPGD very soon!

Thanks very much!